# Profiling of Health-Promoting and Taste-Relevant Compounds in Sixteen Radish (*Raphanus sativus* L.) Genotypes Grown under Controlled Conditions

**DOI:** 10.3390/foods12152823

**Published:** 2023-07-25

**Authors:** Tibor Béres, Nikola Štefelová, Sanja Ćavar Zeljković, Pavel Kopecký

**Affiliations:** 1Czech Advanced Technology and Research Institute, Palacky University, Šlechtitelů 27, 78371 Olomouc, Czech Republic; tibor.beres@upol.cz (T.B.); nikola.stefelova@seznam.cz (N.Š.); 2Centre of the Region Haná for Biotechnological and Agricultural Research, Department of Genetic Resources for Vegetables, Medicinal and Special Plants, Crop Research Institute, Šlechtitelů 29, 78371 Olomouc, Czech Republic; kopeckyp@vurv.cz

**Keywords:** radish, chemical profiling, indoor cultivation, glucosinolates, amino acids, sugars, liquid chromatography

## Abstract

It is becoming increasingly challenging to maintain crop yields and quality as the global climate changes. The aim of this study was to determine whether and how the profile of health-promoting and taste-related compounds of radishes changes within a growing season. A total of 16 radish (*Raphanus sativus* L.) genotypes that are commercially available on the Czech market were assessed by means of chemical analysis. Radishes were cultivated in three independent growing cycles under controlled conditions, and the effects of the genotype and growing cycle, as well as their interactions, on the chemical traits were evaluated. Most of the variability in chemical composition was associated with the growing cycle, which accounted for 51.53% of total variance, followed by the genotype (26% of total variance). The interaction between the growing cycle and genotype explained 22.47% of total variance. The growing cycle had the strongest effect on amino acid profiles. More specifically, the amino acids that are known to contribute to overall taste (glycine, along with glutamic and aspartic acids) showed the highest degree of variation, while the amino acids related to glucosinolate biosynthesis (methionine, isoleucine, tryptophan, and phenylalanine) showed relatively low variability. On the other hand, indole glucosinolates were found to differ the most between genotypes.

## 1. Introduction

Although the consumption of radishes (*Raphanus sativus* L.) in the Czech Republic does not reach the levels typical for Southeast Asia, this root vegetable is experiencing rising popularity, as evidenced by an annual report from the Czech Ministry of Agriculture [1]. The inclusion of radishes in the diet is interesting not only due to their complex sensory characteristics, which include a sharp, pungent, yet sweet taste, but also to the health benefits that are associated with the wide range of biologically active natural substances found in radishes. Cancer and diabetes represent a worldwide problem and are considered “diseases of civilization”. The oncologic burden in the Czech Republic is one of the highest in the world and shows an increasing trend [2]. Moreover, according to the International Diabetes Federation, almost 10 percent of the Czech adult population suffers from diabetes [3].

Many researchers have emphasized the chemo-preventive (anticarcinogenic) and antidiabetic activities of Brassica vegetables, which were recently reviewed in detail by Soundararajan and Kim [4] and Banihani [5]; however, the medicinal value of radishes extends beyond cancer and diabetes. For instance, there is evidence that the consumption of radishes can alleviate stomach disorders, urinary infections, hepatic inflammation, cardiac disorders, and ulcers. The comprehensive nutraceutical potential of radishes is reviewed by Mannivannan et al. [6].

Glucosinolates (GLSs) are phytochemicals typically found in members of the Brassicales order that contribute significantly to biological activity. From the chemical point of view, they represent β-thioglucoside N-hydroxysulfates. Upon mechanical damage that disrupts cell compartmentation, GLSs interact with the enzyme myrosinase. Following enzymatic action, biologically active isothiocyanates, indoles, and nitriles, among others, are formed. When compared with broccoli, which is commonly considered one of the healthiest vegetables, radishes have a higher conversion rate of GLSs to isothiocyanates. Additionally, broccoli cultivation is sensitive to environmental factors and requires professional skills, while it is rather undemanding to grow radishes [7].

Amino acids, particularly alanine, leucine, isoleucine, valine, methionine, phenylalanine, tyrosine, and tryptophan, are biosynthetic precursors of the GLS side chain, with the composition of amino acid moieties significantly affecting the physicochemical properties and biological activity of GLS species [8]. It was also recently shown that some free amino acids can be useful chemical markers for distinguishing various genotypes of basil (*Ocimum basilicum* L.) or individual species of mint (*Mentha* sp.) and onion (*Allium* sp.) [9,10]. Moreover, it was hypothesized that free amino acids also contribute to the overall taste of Brassica vegetables. Whereas valine and leucine are associated with bitterness, alanine, threonine, serine, and proline are known to contribute to sweetness [11].

Apart from the latter, the sweet taste of Brassica vegetables is mainly due to sugars such as glucose, fructose, and sucrose. From the consumer’s point of view, this content is important because it counteracts the perception of bitterness and/or pungency caused by the glucosinolates [11]. Furthermore, sugars play an essential role as a source of energy and carbon for plant development [12].

Increased consumer awareness means that breeders and seed companies are witnessing more demand to characterize different genotypes of vegetables via chemical analysis. The phytochemical composition of a plant affects not only its overall taste, but more importantly, its health-promoting properties. It was recently shown that environmental factors such as rainfall, along with soil and air temperature, influence the quality of the final product [12].

The aim of this study was to determine whether, and how, the profile of health-promoting and taste-related compounds of radishes changes within the season when grown under controlled conditions. To this end, a total of sixteen genotypes of radish available on the Czech market was selected, and the GLS, amino acid, and sugar profiles were determined following cultivation in three independent growing cycles. The respective contributions of the health-promoting and sensorially important metabolites to the overall phytochemical variability were also evaluated and discussed. Understanding the contributions of genetic and environmental factors to overall nutritional value should lead to improvements in product quality by breeders and growers alike. It can be postulated that the availability of radishes of superb quality, both in terms of taste and health benefits, could become an important part of easing the modern health burden associated with cancer and diabetes.

## 2. Materials and Methods

### 2.1. Radish Cultivation, Harvesting, and Processing

Seeds of sixteen radish genotypes were obtained from Moravoseed company (Moravoseed CZ, Mikulov, Czech Republic), with the names and morphological characteristics listed in Table 1. The seeds were sown in 3 × 3 cm planters. The germinated plants were regularly watered with a 1 g L^−1^ solution of Kristalon Start fertilizer (AGRO CS, Rikov, Czech Republic) to achieve a stable watering level of 1 cm. A phytotron (PSI, Drasov, Czech Republic) was used to establish controlled conditions; more specifically, a photoperiod of 16 h day/8 h night with an illumination intensity of 82 μE.m^−2^.s^−1^, measured using an LI-189 quantum/radiometer/photometer handheld device (Lincoln, NE, USA), and temperatures of 23 °C day/15 °C night. Upon reaching maturity (30 to 45 days after seeding), 10 plants from each cultivar were randomly selected, after which the bulbs were harvested, washed, cut into 4 pieces, and freeze-dried for 96 h at a pressure of −0.31 to −0.01 mbar (Christ Beta 1–8 LD plus, Martin Christ Gefriertrocknungsanlagen GmbH, Osterode am Harz, Germany). Following drying, the samples were homogenized in 25 mL stainless-steel vessels with a stainless-steel ball (1 cm diameter) using an oscillating ball mill (Retsch MM400, Retsch, Haan, Germany) at a frequency of 27 Hz for three minutes. The pulverized material was stored at laboratory temperature in sealed centrifuge tubes.

### 2.2. Intact Glucosinolate Analysis

Intact glucosinolates were analyzed via UHPLC-ESI-MS/MS using an UltiMate™ 3000 liquid chromatographic system consisting of binary pumps, an autosampler, and a thermostatted column compartment. The effluent was driven to the HESI ion source of a TSQ Quantum Access Max triple quadrupole mass spectrometer (Thermo Fisher Scientific, Waltham, MA, USA). The chromatographic conditions were adopted from Glauser et al. [13] with modifications. The chromatographic column, an ACQUITY UPLC^®^ CSH C18, 2.1 × 100 mm, 1.7 µm (Waters, Milford, MA, USA), which was kept at 25 °C, was used as the stationary phase, while 0.05% formic acid in water (*v*/*v*) and acetonitrile (*v*/*v*) served as mobile phases A and B, respectively. Optimal separation was achieved with gradient elution at a flow rate of 0.4 mL min^−1^. The analysis started with an isocratic step of 2% B for 0.5 min, followed by a linear gradient to 30% B from 0.5 to 15 min. After a washout step at 90% B from 15.5 to 16.5 min, the column was re-equilibrated to initial conditions, resulting in a total run time of 20 min. The detection was performed in negative electrospray ionization (ESI) mode with the following parameters: spray voltage 3500 V; vaporizer temperature 350 °C; sheath gas pressure 45 psi; auxiliary gas pressure 1 psi; source temperature 320 °C; collision energy 25 V. The GLS standards were purchased from PhytoPlan (Heidelberg, Germany), PhytoLab (Vestenbergsgreuth, Germany), or Extrasynthase (Bourgogne, France), while LC-MS grade water, acetonitrile, methanol, and formic acid were purchased from Merck (Darmstadt, Germany). Glucoraphasatin was purchased as a mixture of E/Z isomers (76.01% of E-isomer), and this was taken into account during calibration. The multiple reaction monitoring (MRM) transitions are described in Table 2.

As for sample preparation, the protocol published by Lee et al. [14] was adopted with minor modifications. Briefly, 25 mg of pulverized material was weighed into 2 mL Eppendorf tubes in three independent replicates and sonicated with 1 mL of 75% *v*/*v* methanol containing 0.1% formic acid for 15 min. Following centrifugation, 14,500× *g* for 10 min, 100 µL of the supernatant was diluted 10-fold into water, and the internal standard (sinigrin, IS) was added to a final concentration of 10 µmol L^−1^. After subsequent filtration through a 0.22 µm nylon syringe filter (Chromservis, Prague, Czech Republic), a 5 µL sample was injected into the UHPLC-ESI-MS/MS system.

### 2.3. Free Amino Acid Analysis

The levels of free amino acids were determined according to the method presented by Abdelhakim et al. [15], with minor modifications. Briefly, 50 mg of homogenized material was extracted with 1 mL 50% EtOH containing 1 μM isotope-labeled amino acids (leucine, alanine, proline, glutamine, and glutamate). After 10 min of sonication, extracts were centrifuged at 14,500× *g* for 5 min, and 250 μL of the supernatant was transferred into a new tube and evaporated to dryness. The residue was dissolved in 50 μL of mobile phase consisting of 20 mM ammonium formate, pH 3.0 (mobile phase A), and 0.2% formic acid in ACN (mobile phase B). UHPLC-MS/MS analysis was performed on a Nexera X2 UHPLC (Shimadzu Handels, Korneuburg, Austria) coupled with an MS-8050 (Shimadzu Handels). Chromatographic separation was performed on an Acquity UPLC BEH AMIDE (50 × 2.1 mm; 1.7 μm particle size) column with a corresponding pre-column. The employed analytical conditions are described in a previously published study [15].

### 2.4. Sugar Analysis

Sugar analysis was conducted with HPLC-ELSD according to a protocol described in O’Donoghue et al. [16], with minor modifications. Briefly, 20 mg of pulverized material was extracted in three independent replicates with 2 mL of 75% methanol (*v*/*v*) for 15 min. After centrifugation at 14,500× *g* for 10 min, 750 µL of the supernatant was completely evaporated using a centrifugal vacuum concentrator (Labconco, Kansas City, MO, USA) at 40 °C. Upon reconstitution in 750 µL of water and subsequent filtration through a 0.22 µm nylon syringe filter (Chromservis), 10 µL of the sample was injected on a WATREX polymer IEX Ca-form 8 × 300 mm, 8 µm particle size column (WATREX, Prague, Czech Republic). The Smartline HPLC system (Knauer, Berlin, Germany) was coupled with an Alltech 3300 ELSD detector (Thermo Fisher). Sucrose, glucose, and fructose were identified and quantified using authentic standards (Lach-Ner, Neratovice, Czech Republic). MilliQ water (Milipore Sigma, Billerica, MA, USA), at a flow rate of 0.5 mL min^−1^, was used as the mobile phase. The ELSD detector operated at a temperature of 80 °C and a nitrogen flow rate of 2 L h^−1^. The analysis had a total run time of 20 min.

### 2.5. Statistical Analysis

The statistical analyses were performed in RStudio (R Software version 4.1.0). Any non-detected values were replaced with 2/3 of the minimum detected value for the respective variable. The missing values for ‘Faraon’ and ‘Ester’ for cycle 2021 were imputed based on the values for the other genotypes’ values in the same cycle and the rank of ‘Faraon’ and ‘Ester’ among the genotypes in the other two cycles (for the given variable). The data were log-transformed and visualized via boxplots. Spearman correlation coefficients were computed and displayed. Their significance was investigated via correlation tests including the Benjamini–Yekutieli correction. The effects of the cycle and genotype were examined via ANOVA simultaneous component analysis (ASCA). That is, first, the data matrix was decomposed into three matrices: one corresponding to the factor growing cycle, one to the factor genotype, and one to their interaction. Next, two principal component analyses (PCAs) were performed (one after subtracting the overall genotype effect and one after subtracting the overall growing cycle effect), and the respective biplots were constructed.

## 3. Results and Discussion

### 3.1. Genotype Characteristics and Variables

This study was conducted on 16 radish (*Raphanus sativus* L.) genotypes that are commercially available in the Czech Republic. The basic morphological characteristics of these radish genotypes are described in Table 1. Of the 16 genotypes, 13 have bulbs of a circular shape, while the bulbs of the genotypes ‘Slovana’, ‘Rampouch’, and ‘Felicia’ are cylindric. The color of the bulbs of most genotypes varies from red to purple, while three genotypes produce white radishes (‘Rampouch’, ‘Albena’, and ‘SU VB’) and one genotype produces yellow radishes (‘Zlata‘). Most of the genotypes produce radishes with white to off-white flesh, while the flesh of radishes representing the ‘Ester’, ‘Ria’, Lidka’, and ‘Poloneza’ genotypes also have pink tones. The growing season for radishes is generally short; the genotypes ‘Faraon’, ‘Ester’, and ‘Ria’ produce the earliest vegetables, while the genotype ‘Zlata’ is characterized by a mid–late harvest and the genotype ‘Viola’ has a late harvest. Most of the genotypes do not produce radishes with a tendency to be spongy; the exceptions are ‘Ria’, ‘Slovana’, ‘Albena’, and ‘Felicia’ (Table 1).

### 3.2. Intact Glucosinolate Content

A large share of the health benefits of Brassica vegetables is attributed to GLSs [6]. In a publication by Yi et al. [17], 8 distinct GLSs were identified from the total of 15 GLSs reported in [18]. We have identified 7 intact GLSs using ESI-LC-MS/MS in bulbs of 16 radish genotypes. The aliphatic GLS species include GRA, GRE, GRH, GAS, and GER, whereas GBC and 4-MGBS represent indole GLSs (Table 2). In the current work, the levels of intact GLSs were determined; this is in contrast with the work of Yi et al., which analyzed the levels of desulfo-GLS [17]. Nevertheless, Lee et al. stated that both analyses of desulfo-GLS and intact GLS provide similar estimates of GLS concentrations [14].

According to previous publications, GRH is the most abundant GLS in radishes [12,17]. Indeed, GRH was the predominant GLS in each of the studied genotypes, accounting—on average—for over 90% of the total GLS (Appendix A). Although a complete chromatographic separation of the E/Z isomers was achieved, the Z-isomer was not detected in any of the radish samples. The highest observed concentrations of GRH ranged from 0.38 µmol/g DW in the genotype ‘Stela’ in growing cycle 2021 to 34 µmol/g DW in the genotype ‘Albena’ in cycle 2022_2. These values generally agree with what was reported by Yi et al. [17] despite the differences in analytical approaches.

According to Bell et al. [11], raphasatin, which originates from GRH, contributes to the overall pungent taste of radishes. Thus, consumers who prefer a mild taste would want to select the genotype ‘Rampouch’, as the levels of GRH reached only 4.1, 6.3, and 11.5 µmol/g DW during the three growing cycles (Figure 1). In contrast, the genotypes ‘Albena’ and ‘Lidka’ are more suitable for consumers who demand a more pungent radish, as the contents of GRH ranged from 25.8 to 34.1 µmol/g DW in ‘Albena’ and from 22.8 to 25.4 µmol/g DW in ‘Lidka’ (Appendix A, Figure 1).

The second most abundant GLS was GRE, but the measured concentrations only reached low micromolar levels (Appendix A). It was recently found that the conversion of GRH to GRE is catalyzed by the enzyme FMOGS-OX1 during the transition from the vegetative phase to flowering [19]. The rate of conversion sharply increases during flowering, and the ratio of GRH to GRE in seeds is completely reversed six weeks after pollination. The dynamics of the GRH to GRE conversion could therefore serve as a potential marker of bolting, which is an important trait when producing commercial radish cultivars. The genotypes ‘Diana’ and ‘Vera’ showed the highest proportion of GRE (Figure 1, Appendix A); in ‘Diana’, GRE accounted for more than 30% of the total GLS in growing cycles 2022_1 and 2022_2. In radishes, bolting can be indicative of elevated soil temperature (or other types of stress), which suggests that ‘Diana’ might not be the most appropriate genotype to select for the weather extremes that are associated with global climate change.

Although indole GLSs are mainly associated with superior turnip accessions, these compounds have also been identified in some radish genotypes; indole GLSs have been proposed as chemical markers for distinguishing turnip and radish genotypes when morphological evaluation alone is insufficient [17]. In the studied genotypes, indole GLSs did not exceed 5% of the total GLSs. The highest levels of GBC were found in the genotype ‘Lidka’ and ranged from 26.2 nmol/g DW in growing cycle 2022_2 to 97.7 nmol/g DW in 2021. This genotype also contained noticeable levels of 4-MGBS, as shown in Figure 1 and Appendix A, which may indicate turnip ancestry. In contrast, neither GBC nor 4-MGBS were detected in ‘Rampouch’ radishes across any of the growing seasons, which could explain the lack of pungency and bitterness, generally associated with indole GLSs, in this genotype.

### 3.3. Free Amino Acid Content

The free amino acid composition of vegetables has a large effect on the overall sensory perception of the product. Bell et al. [11] recently reviewed the associations between various compounds and taste perception among Brassicaceae vegetables, with free amino acids shown to be important to the perceived taste. Moreover, it was shown that free amino acids have taxonomic value; i.e., these metabolites can be used to discriminate between close relatives, such as onions and shallots (*Allium cepa* and *Allium cepa* group aggregatum L.) [10] or even genotypes of a single species, e.g., salad (*Lactuca sativa* L.) [20], mint (*Mentha* sp. L.), and basil (*Ocimum basilicum* L.) [9].

A chemical composition study, including sensory analysis, conducted on rocket (*Eruca sativa* L.) reported that higher overall levels of free amino acids cause rocket leaves to be less bitter and pungent [21]. More specifically, glycine, alanine, threonine, serine, and proline are associated with a sweet taste. In this study, the radish genotype ‘Ester’ contained high levels of serine and proline, with the genotype ‘Lidka’ also showing high levels of proline. On the other hand, no genotype was found to be rich in glycine, alanine, or threonine (Figure 1, Appendix A). Aspartic and glutamic acids are associated with a savory taste, which is often referred to as umami. The boxplots show that the genotype ‘Viola’ had levels of aspartic and glutamic acids that exceeded the median values across all three growing cycles (Figure 1). In contrast, the genotype ‘Vera’ showed low levels, i.e., below median values, of these two flavor-enhancing amino acids across the growing cycles. Valine, leucine, isoleucine, and methionine have been described to contribute to a bitter taste. Notably, the genotypes ‘Ester’, ‘Ria’, and ‘Lidka’ demonstrated relatively high valine and leucine contents (Figure 1). Moreover, the highest isoleucine levels were observed in the ‘Faraon’ radishes, followed by the ‘Lidka’ genotype; both showed levels that were consistently above the median values. Tryptophan also contributes to a bitter taste and is a precursor of indole GLSs. The highest tryptophan concentrations were observed in the ‘Ester’, ‘Ria’, ‘Faraon’, and ‘Viola’ genotypes, while ‘Diana’ radishes showed the lowest values.

### 3.4. Free Sugar Content

The sugar profile of a vegetable can significantly modulate the overall taste perception by reducing bitterness and influencing aftertaste. It was previously hypothesized that the taste of a Brassica vegetable can be modified through changes in the sugar–GLS ratio [11]. Sucrose, glucose, and fructose were detected in the radish bulbs. The values obtained for the sixteen genotypes across three growing cycles are summarized in Table 3. Sucrose was the least abundant saccharide, with median values ranging from 37.99 mg/g DW in 2021 to 46.98 mg/g DW in the 2022_1 cycle; a minimum value of 8.01 mg/g DW was observed in the genotype ‘Poloneza’ during growing cycle 2021, while a maximum value of 101.02 mg/g DW was observed in the genotype ‘Faraon’ during the 2022_1 cycle. Glucose was the predominant saccharide, with median values ranging from 274.65 mg/g DW in the 2022_2 cycle to 306.77 mg/g DW in cycle 2022_1. A minimum value of 169.53 mg/g DW was observed in the genotype ‘Faraon’ during the 2022_1 cycle, while a maximum value of 306.77 mg/g DW was observed in the genotype ‘Rampouch’ during the same cycle. The median fructose content values were the lowest in 2021 (152.02 mg/g DW) and the highest in 2022_1 (190.12 mg/g DW). The genotype ‘Ester’ demonstrated the lowest fructose content (105.1 mg/g DW) in 2021, while a maximum value of 228.44 mg/g DW was observed in the genotype ‘Lada’ during the growing cycle 2022_1. These free sugar contents are slightly higher than what was reported in radish accessions by Kang et al. [22]; these discrepancies can be explained by differences in the analytical methods and the studied genotypes. For instance, Kang et al. used accessions from Korean, German, and Japanese gene banks, while the genotypes analyzed in the present study are commercial cultivars that have been adapted to the taste preferences of Czech consumers.

### 3.5. Relationships between Metabolites

Elucidation of the relationships between different metabolites is important, as it can help breeders to improve specific traits such as the content of health-promoting or taste-related compounds. Spearman correlation coefficients were calculated to investigate the possible relationships between the metabolites quantified from 16 radish genotypes (Figure 2). Correlation coefficients above 0.6 are regarded as strong; the complete list of values is in Appendix A. In general, positive correlations were found between different amino acids, indicating shared biosynthetic pathways. Similar results were obtained by Park et al. [23]. Isoleucine was the exception, showing a pronounced inverse correlation with serine, followed by alanine, hydroxyproline, threonine, glycine, and glutamic and aspartic acids (Figure 2). Positive correlations were also observed within the group of aliphatic GLSs (Figure 2). As for amino acid–GLS relationships, tryptophan showed a negative correlation with GRA, as did lysine with both GRA and GAS. No significant correlation was found between aliphatic GLSs and methionine or between indole GLSs and tryptophan. These results are in line with the ones of Rosa et Gomes and Liu et al. finding no significant correlation of glucosinolates with their precursor amino acids in various Brassica vegetables [24,25]. This indicates other important roles of amino acids besides GLS biosynthesis, as suggested by Rosa et Gomes [24]. On the other hand, Liu et al. found positive correlations between the levels of the indole GLS 4MGBS and amino acids LEU and ILE [25]. No such relationship was observed in the current work. Instead, 4MGBS was positively correlated with the amino acids tyrosine, alanine, glutamine, serine, asparagine, arginine, lysine, and glutamic and aspartic acids. The differences can be attributed to the different Brassica species studied and to differences in the analytical approach.

### 3.6. The Effects of Growing Cycle and Genotype on Detected Levels of Metabolites

Based on data matrix decomposition, the largest share of data variability was attributed to the growing cycle (which explains 51.53% of the total variance), followed by the genotype (26% of total variance) and interactions between the growing cycle and genotype (22.47% of total variance). The impact of the growing cycle is visualized in Figure 3. Growing cycle 2022_2 noticeably differed from 2021 and 2022_1. In particular, it was characterized by low levels of free amino acids, especially glycine and glutamic and aspartic acids. No genotype deviates from this trend. This clear difference can probably be attributed to the fact that while growing cycle 2022_2 was conducted in autumn months (September–October), cycles 2021 and 2022_1 were conducted in winter months (January–February).

The role of the genotype is presented in Figure 4. GLSs were identified as the major contributors to the distinction between the genotypes. The indole GLSs (GBS and 4MGBS) show particularly high variability among the genotypes. The genotypes ‘Felicia’, ‘Lidka’, and ‘Diana’ were characterized by relatively high levels of GLSs throughout the growing cycles with respect to the other genotypes, while the genotype ‘Rampouch’ showed the lowest abundance of GLSs across each of the growing cycles. Some genotypes showed a fair degree of variation in terms of metabolite levels, especially free amino acids, throughout the growing cycles. For instance, considerable fluctuation was observed in the genotype ‘Albena’, as these radishes had relatively high values of many amino acids in one growing cycle (2021), but relatively low values in another (2022_1). The ‘Rampouch’ and ‘Lada’ genotypes also showed substantial shifts in terms of free amino acids across the growing cycles.

The significant difference in amino acid levels between growing cycle 2022_2 and growing cycles 2021 and 2022_1 could be caused by differences in seed storage time. Attri et al. suggested that seed germination percentage (and presumably also the metabolite content at harvest time) could be affected by seed storage time [26]. The effect of storage time on metabolite levels needs to be evaluated further by performing additional growing cycles.

## 4. Conclusions

A total of 30 metabolites related to important chemical/sensorial traits were studied in 16 commercial radish genotypes across 3 independent growing cycles conducted under controlled conditions. Subsequent data analysis revealed that amino acids, particularly glycine, along with aspartic and glutamic acids, showed the most variation across the three growing cycles. As for GLSs, the indole compounds, 4MGBS, and GBC showed the highest degree of variability across the radish genotypes. Sugars showed the lowest degree of variability. Surprisingly, environmental factors were found to exert the largest impact on metabolite variance; however, the effect of seed storage time cannot be ruled out. This indicates that environmental factors play an important role in the overall sensorial quality of radishes. Future research could help identify which factors have the largest impact on the quality of radishes, which would be pivotal to attracting further interest for this healthy vegetable.

## Figures and Tables

**Figure 1 foods-12-02823-f001:**
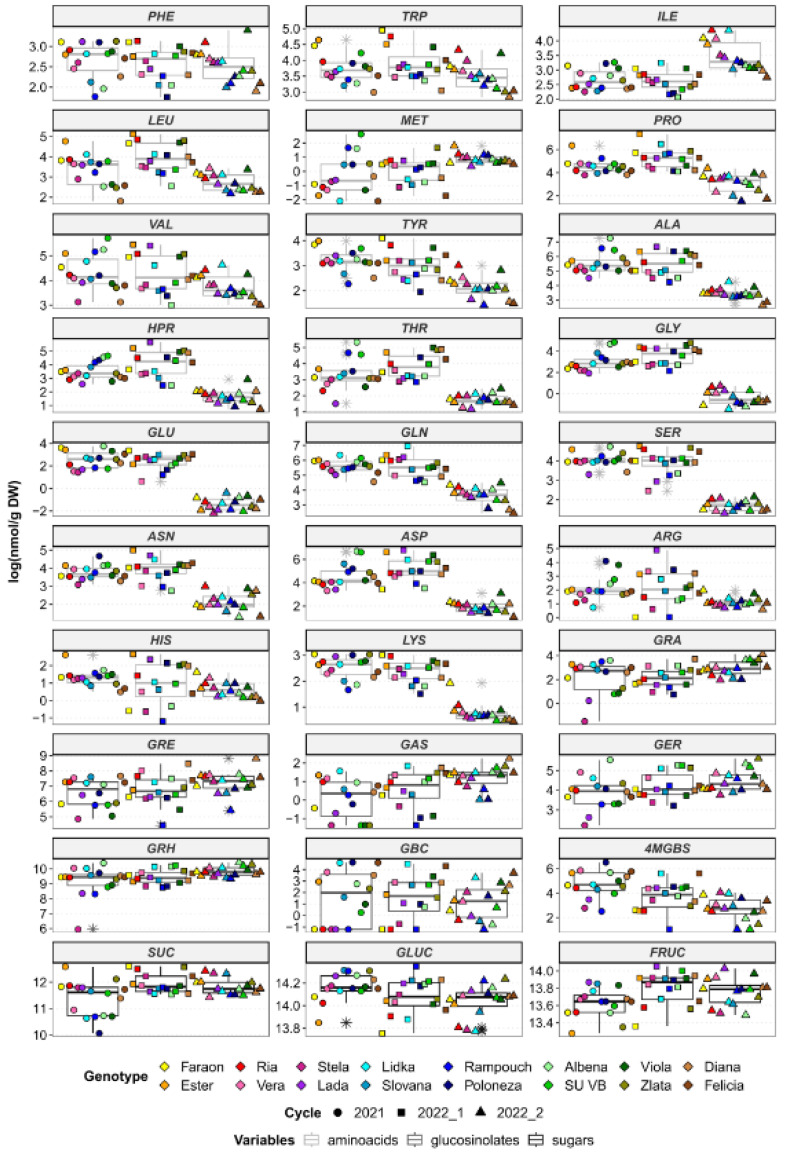
Boxplots illustrating the abundances of the studied metabolites (in log-scale). PHE—phenylalanine, TRP—tryptophan, ILE—isoleucine, LEU—leucine, MET—methionine, PRO—proline, VAL—valine, TYR—tyrosine, ALA—alanine, HPR—hydroxyproline, THR—threonine, GLY—glycine, GLU—glutamic acid, GLN—glutamine, SER—serine, ASN—asparagine, ASP—aspartic acid, ARG—arginine, HIS—histidine, LYS—lysine, GRA—glucoraphanin, GRE—glucoraphenin, GAS—glucoalyssin, GER—glucoerucin, GRH—glucoraphasatin, GBC—glucobrassicin, 4MGBS—4-Methoxyglucobrassicin, SUC—sucrose, GLUC—glucose, FRUC—fructose. The tested genotypes are marked with different colors, while the three controlled growing cycles are distinguished as circles (2021), squares (2022_1), or triangles (2022_2).

**Figure 2 foods-12-02823-f002:**
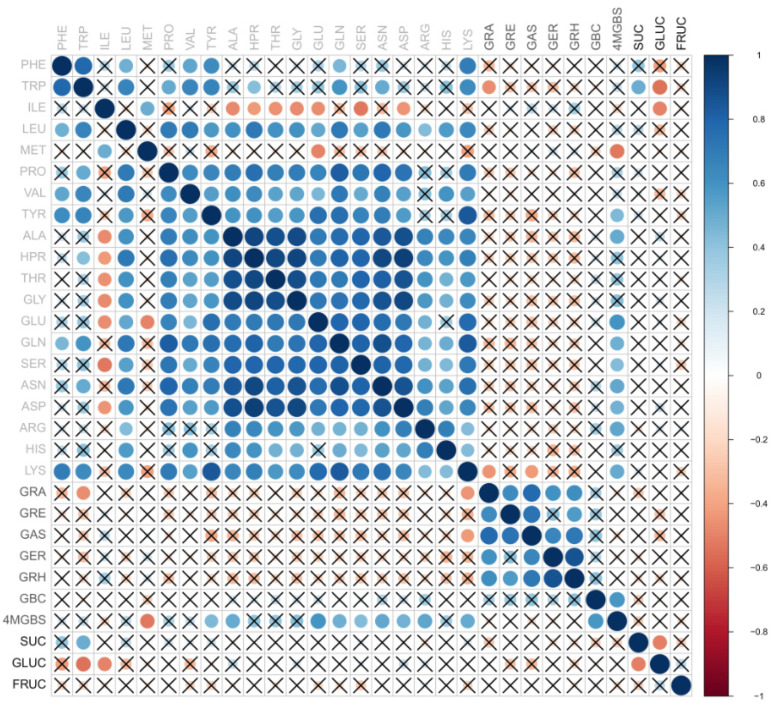
Correlogram displaying the Spearman correlation coefficients between various detected radish metabolites. The correlation coefficients are proportional to the dot area and color. The insignificant values (according to correlation tests, including Benjamini–Yekutieli correction) are crossed out. The features are not sorted along hierarchical clustering. PHE—phenylalanine, TRP—tryptophan, ILE—isoleucine, LEU—leucine, MET—methionine, PRO—proline, VAL—valine, TYR—tyrosine, ALA—alanine, HPR—hydroxyproline, THR—threonine, GLY—glycine, GLU—glutamic acid, GLN—glutamine, SER—serine, ASN—asparagine, ASP—aspartic acid, ARG—arginine, HIS—histidine, LYS—lysine, GRA—glucoraphanin, GRE—glucoraphenin, GAS—glucoalyssin, GER—glucoerucin, GRH—glucoraphasatin, GBC—glucobrassicin, 4MGBS—4-Methoxyglucobrassicin, SUC—sucrose, GLUC—glucose, FRUC—fructose.

**Figure 3 foods-12-02823-f003:**
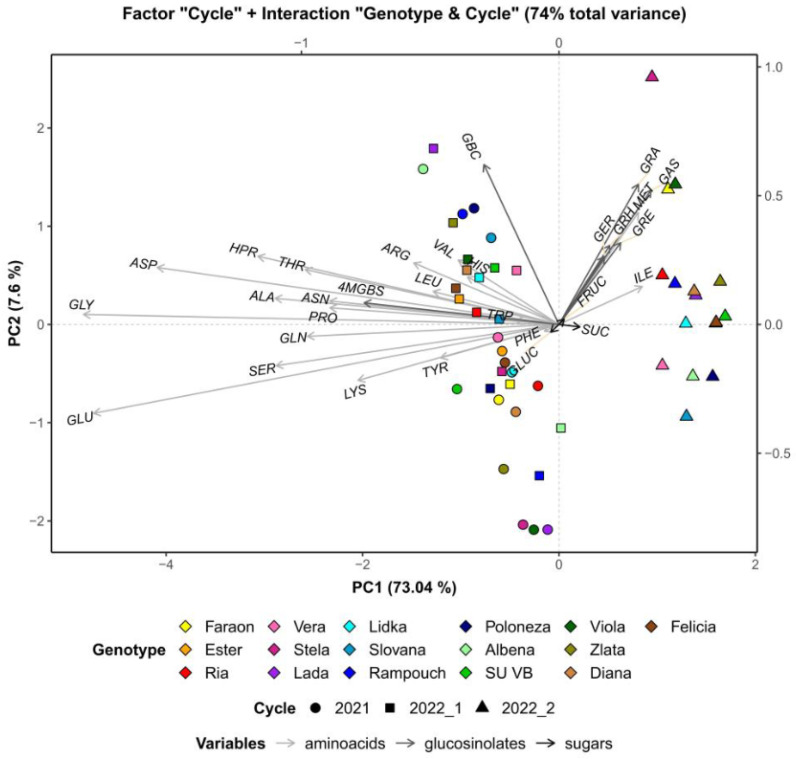
PCA biplot of radish metabolites after subtracting the overall genotype effect. PHE—phenylalanine, TRP—tryptophan, ILE—isoleucine, LEU—leucine, MET—methionine, PRO—proline, VAL—valine, TYR—tyrosine, ALA—alanine, HPR—hydroxyproline, THR—threonine, GLY—glycine, GLU—glutamic acid, GLN—glutamine, SER—serine, ASN—asparagine, ASP—aspartic acid, ARG—arginine, HIS—histidine, LYS—lysine, GRA—glucoraphanin, GRE—glucoraphenin, GAS—glucoalyssin, GER—glucoerucin, GRH—glucoraphasatin, GBC—glucobrassicin, 4MGBS—4-Methoxyglucobrassicin, SUC—sucrose, GLUC—glucose, FRUC—fructose. Genotypes are marked with different colors, while the growing cycles can be distinguished as circles (2021), squares (2022_1), or triangles (2022_2).

**Figure 4 foods-12-02823-f004:**
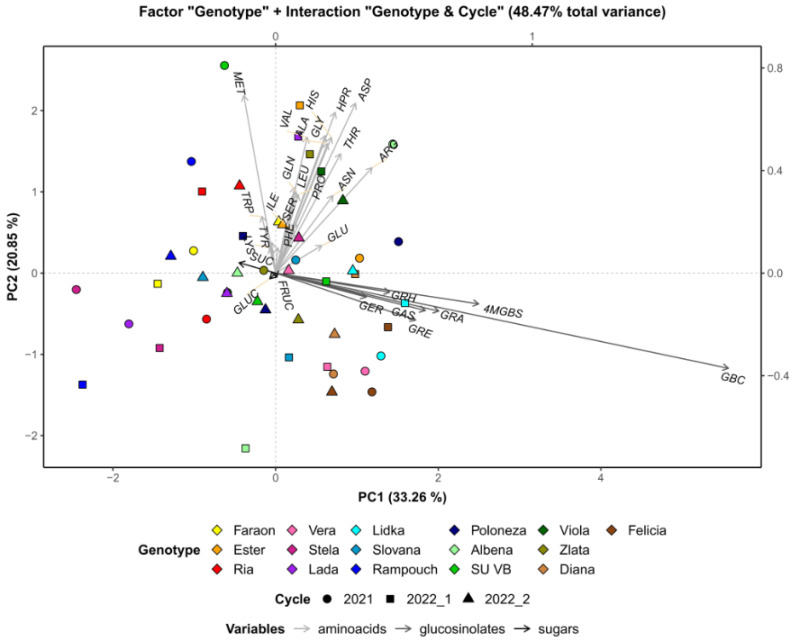
PCA biplot of radish metabolites after subtracting the overall growing cycle effect. PHE—phenylalanine, TRP—tryptophan, ILE—isoleucine, LEU—leucine, MET—methionine, PRO—proline, VAL—valine, TYR—tyrosine, ALA—alanine, HPR—hydroxyproline, THR—threonine, GLY—glycine, GLU—glutamic acid, GLN—glutamine, SER—serine, ASN—asparagine, ASP—aspartic acid, ARG—arginine, HIS—histidine, LYS—lysine, GRA—glucoraphanin, GRE—glucoraphenin, GAS—glucoalyssin, GER—glucoerucin, GRH—glucoraphasatin, GBC—glucobrassicin, 4MGBS—4-Methoxyglucobrassicin, SUC—sucrose, GLUC—glucose, FRUC—fructose. Genotypes are marked with different colors, while growing cycles can be distinguished as circles (2021), squares (2022_1), or triangles (2022_2).

**Table 1 foods-12-02823-t001:** Basic characteristics of the radish genotypes used in this study.

Genotype	Bulb Shape	Bulb Color	Flesh Color	Days from Sowing to Maturity	Tendency to be Spongy in Late Maturity
Faraon	spheric	red	off-white	28–32	no
Ester	spheric	bright red	off-white-pink	28–32	no
Ria	spheric	bright red	off-white-pink	30–32	yes
Vera	spheric	red	white	30–33	NA
Stela	spheric	red	white	32–35	no
Lada	spheric	dark red	white	33–37	no
Lidka	spheric	red	off-white-pink	35–39	no
Slovana	cylindric	red-white	off-white	32–35	yes
Rampouch	cylindric	white	white	35–38	NA
Poloneza	spheric	red-white	off-white-pink	29–33	no
Albena	spheric	white	white	30–34	yes
SU VB	spheric	white	NA	NA	NA
Viola	spheric	purple	off-white	35–38	no
Zlata	spheric	yellow	off-white	36–40	no
Diana	spheric	purple-white	off-white	30–35	no
Felicia	cylindric	purple white	white	32–37	yes

NA—not available.

**Table 2 foods-12-02823-t002:** Common names, abbreviations, retention times and MRM transitions of the analyzed glucosinolates.

Glucosinolate	Abbreviation	Retention Time	MRM Transitions
Glucoraphanin	GRA	4.28	436 > 421, 372
Glucoraphenin	GRE	4.52	434 > 419, 275
Sinigrin (IS)	SIN	4.97	358 > 97, 75
Glucoalyssin	GAS	5.33	450 > 435, 386
Glucoerucin	GER	9.92	420 > 96, 75
E-Glucoraphasatin	GRH	10.59	418 > 96, 75
Glucobrassicin	GBC	11.47	447 > 96, 75
4-Methoxyglucobrassicin	4-MGBS	12.88	477 > 96, 75

**Table 3 foods-12-02823-t003:** Concentrations of free sugars obtained via HPLC-ELSD analysis of sixteen radish genotypes across three growing cycles. The values are the mean value of three replicates (±SD) and are reported in mg/g DW. SUC—sucrose, GLUC—glucose, FRUC—fructose.

Cycle	2021			2022-1			2022-2		
Analyte/ Genotype	SUC	GLUC	FRUC	SUC	GLUC	FRUC	SUC	GLUC	FRUC
Faraon	39.32 ± 0.83	253.00 ± 4.12	116.87 ± 2.26	101.02 ± 7.55	169.53 ± 16.81	114.11 ± 14.55	55.07 ± 3.58	228.81 ± 14.36	176.20 ± 14.03
Ester	99.48 ± 0.92	186.14 ± 1.40	105.10 ± 1.40	51.76 ± 3.68	225.87 ± 10.54	199.60 ± 9.73	56.02 ± 3.13	219.22 ± 8.55	197.19 ± 5.26
Ria	48.54 ± 0.71	221.67 ±2.37	134.18 ± 1.87	91.38 ± 2.03	197.26 ± 4.25	141.98 ± 2.31	84.96 ± 2.17	178.71 ± 2.50	136.91 ± 4.10
Věra	19.43 ± 1.07	251.66 ± 2.61	155.75 ± 0.66	48.74 ± 0.91	223.80 ± 2.35	185.17 ± 1.80	31.25 ± 0.66	236.04 ± 8.78	201.23 ± 10.41
Stela	45.36 ± 0.73	258.17 ± 3.82	160.88 ± 3.03	69.66 ± 2.93	239.18 ± 9.86	200.36 ± 10.49	78.00 ± 5.97	174.81 ± 5.31	132.91 ± 4.20
Lada	45.28 ± 0.85	281.71 ± 1.62	189.95 ± 2.38	38.73 ± 1.11	264.03 ± 6.68	228.44 ± 3.83	43.04 ± 3.17	246.44 ± 6.85	222.91 ± 5.14
Lidka	14.18 ± 0.33	247.08 ± 0.73	171.60 ± 2.10	35.85 ± 1.05	191.84 ± 6.12	156.50 ± 4.81	38.99 ± 2.00	172.92 ± 8.49	151.38 ± 5.65
Slovana	39.35 ± 0.47	296.47 ± 3.71	186.92 ± 5.8	43.71 ± 5.86	270.68 ± 9.88	197.78 ± 11.64	76.95 ± 3.01	204.88 ± 13.05	147.46 ± 10.10
Rampouch	15.02 ± 0.57	294.43 ± 10.07	151.94 ± 5.74	45.81 ± 3.51	306.77 ± 14.06	176.39 ± 8.03	34.32 ± 2.11	270.12 ± 7.07	171.87 ± 4.38
Poloneza	8.01 ± 0.65	252.30 ± 6.52	152.10 ± 3.21	43.38 ± 1.09	257.15 ± 0.61	195.08 ± 1.40	34.58 ± 1.43	241.19 ± 9.99	175.28 ± 8.06
Albena	15.60 ± 1.45	283.37 ± 3.15	133.42 ± 1.58	35.07 ± 2.83	261.78 ± 9.00	179.50 ± 5.63	42.85 ± 3.73	253.69 ± 16.54	129.92 ± 5.05
SU VB	36.66 ± 3.62	246.63 ± 23.03	144.40 ± 12.35	39.15 ± 2.14	268.74 ± 23.21	216.84 ± 17.07	33.73 ± 2.92	232.24 ± 13.97	174.89 ± 9.51
Viola	15.22 ± 0.59	256.06 ± 10.65	183.58 ± 2.65	48.15 ± 8.99	219.37 ± 20.78	198.43 ± 17.09	49.06 ± 2.42	228.68 ± 4.77	209.89 ± 2.05
Zlata	62.39 ± 1.00	294.96 ± 5.50	112.96 ± 1.91	98.41 ± 4.34	229.73 ± 7.65	139.64 ± 3.89	55.16 ± 7.82	274.65 ± 11.02	159.77 ± 5.37
Diana	30.11 ± 2.83	273.08 ± 7.69	156.63 ± 4.79	36.08 ± 2.37	272.76 ± 9.45	205.06 ± 5.50	37.96 ± 3.28	234.63 ± 19.38	178.28 ± 10.33
Felicia	41.91 ± 2.80	261.61 ± 16.73	134.57 ± 10.05	69.91 ± 3.07	208.52 ± 4.69	152.62 ± 6.68	43.64 ± 2.00	236.50 ± 2.98	178.91 ± 3.68
median	37.99	257.12	152.02	46.98	234.46	190.12	43.34	233.43	175.08
max	99.48	296.47	189.95	101.02	306.77	228.44	84.96	274.65	222.91
min	8.01	186.14	105.10	35.07	169.53	114.11	31.25	172.92	129.92

## Data Availability

Please contact the corresponding author.

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
