# Peer review of "Profiling of Health-Promoting and Taste-Relevant Compounds in Sixteen Radish (Raphanus sativus L.) Genotypes Grown under Controlled Conditions"

_foods, 2023, doi:10.3390/foods12152823_

Round 1
Reviewer 1 Report
Comments and Suggestions for Authors
It is increasingly challenging to maintain crop yields and quality in the face of global climate change. Radish is gaining popularity in the Czech Republic. The inclusion of radishes in the diet is not only due to their interesting sensory characteristics but also their associated health benefits. With increased consumer awareness, breeders and seed companies are facing growing demand for the characterization of different radish genotypes through chemical analysis. Thus, the aim of this study was to investigate whether and how the profile of health-promoting and taste-related compounds in radishes changes throughout the growing season. Several interesting results were obtained, which will provide valuable references for improving quality through breeding or cultivation techniques. Moreover, the paper is well-organized and clearly stated.
There are a few minor issues that need attention:
1. In Table 3, it would be good to include standard deviation (SD) values since the reported values are the means of three replicates.
2. It is worth considering if the journal requires the combination of results with the discussion section.
3. In the section "3.5 Relationships between metabolites," while the results are presented, there is a lack of interpretation. It would be helpful to provide an explanation of the significance of conducting this analysis, as it is currently not clear.
Author Response
Dear Reviewer 1,
thank you for your valuable time spent by reviewing our manuscript and for the suggestions.
The responses are listed in the attached file.

Reviewer 2 Report
Comments and Suggestions for Authors
Dear Authors,
Thank you for the ability to review your paper. I think your work is interesting and I especially appreciate the use of a large array authentic standards as well as using analytical methods fine-tuned for each compound class. To improve the paper, please address the following.
Major issues
1., Given non-normality (due to imputting, and otherwise) perhaps using Spearman's rho as a correlation metric would be more robust. Only discuss correlations if they pass a statistical test a p-value adjustment after statistical test (BY correction is to be preferred due to multi-correlating data). See R functions cor.test() p.adjust(). Define "weak" and "strong" correlations.
2., You used a phytotron to grow your plants, but the effect of `growing season` is clearly very significant. How come the growing cycles be that different? Was initial growing been done in the phytotron only? What parameters were not controlled that might have led to this variance? Did you use more than one production batches of seeds for starting the expierments?
3., Irradiance influences radish taproot development and taproot / shoot ratios (Zha and Liu 2018). The literature is mixed regarding illumination requirements, but some authors used as much as 900 μmol m−2 s−1. How was light intensity measured? Provide data on taproot / shoot ratio or taproot plant-1.
4., The discussion needs to be extended by comparing your results (e.g. in the form of RSD / CV values) to those of other researchers on radish chemical variability. Large-scale screening studies include, but are not limited to Malik et al. 2010; Hara et al. 2011.
Minor issues
Moravoseed includes the number of days from sowing to maturity in their description. Add it to Table 1.
E-raphasatin: which vendor provides a stereochemically pure form? Was E and Z detected/separated in your method? If so, please provide details. I am unsure about the possibility of conversion during extraction.
Fig.1. It is impossible (or very hard) to estimate ranks of different cultivars. Sometimes the effect of `year` is clear, sometimes it is not that apparent. A barplot with each accession average + stdev shown for each year separately would make much more sense.
If possible, support your claims in section 3 by citing papers that did taste panel evaluations of radish (or turnip).
Fig.2. In the caption, indicate that features are NOT sorted along hierarchical clustering, as that is typical for correlation heatmaps.
Best regards.
References
Zha L, Liu W (2018) Effects of light quality, light intensity, and photoperiod on growth and yield of cherry radish grown under red plus blue LEDs. Horticulture Environment and Biotechnology 59:511–518. https://doi.org/10.1007/s13580-018-0048-5
Malik MS, Riley MB, Norsworthy JK, Bridges W (2010) Variation of glucosinolates in wild radish (raphanus raphanistrum) accessions. Journal of Agricultural and Food Chemistry 58:11626–11632. https://doi.org/10.1021/jf102809b
Hara M, Torazawa D, Asai T, Takahashi I (2011) Variations in the soluble sugar and organic acid contents in radish (Raphanus sativus L.) cultivars. International Journal of Food Science and Technology 46:2387–2392. https://doi.org/10.1111/j.1365-2621.2011.02761.x
Author Response
Dear Reviewer 2,
thank you for your valuable time spent by reviewing our manuscript as well as for the helpfull suggestions.
The responses are listed in the attached file.
